# Updates to the zoonotic niche map of Ebola virus disease in Africa

**David M Pigott[1,2]\*, Anoushka I Millear[1], Lucas Earl[1], Chloe Morozoff[1]\*, Barbara A Han[3], Freya M Shearer[2], Daniel J Weiss[4,5], Oliver J Brady[2], Moritz UG Kraemer[4,5], Catherine L Moyes[2], Samir Bhatt[4,5], Peter W Gething[4,5], Nick Golding[2,6], Simon I Hay[1,2]**

[1]Institute for Health Metrics and Evaluation, University of Washington, Seattle, United States; [2]Oxford Big Data Institute, Li Ka Shing Centre for Health Information and Discovery, Oxford, United Kingdom; [3]Cary Institute of Ecosystem Studies, New York, United States; [4]Spatial Ecology and Epidemiology Group, University of Oxford, Oxford, United Kingdom; [5]Department of Zoology, University of Oxford, Oxford, United Kingdom; [6]Department of BioSciences, University of Melbourne, Parkville, Australia

**Abstract** As the outbreak of Ebola virus disease (EVD) in West Africa is now contained, attention is turning from control to future outbreak prediction and prevention. Building on a previously published zoonotic niche map (*Pigott et al., 2014*), this study incorporates new human and animal occurrence data and expands upon the way in which potential bat EVD reservoir species are incorporated. This update demonstrates the potential for incorporating and updating data used to generate the predicted suitability map. A new data portal for sharing such maps is discussed. This output represents the most up-to-date estimate of the extent of EVD zoonotic risk in Africa. These maps can assist in strengthening surveillance and response capacity to contain viral haemorrhagic fevers.

**\*For correspondence:** pigottdm@uw.edu (DMP); chloemor@uw.edu (CM)

## Introduction

Since the index case in 2013, the West African Ebola epidemic has killed more than 11,000 people (*World Heath Organization, 2016*) and exposed national and international inadequacies in pandemic preparedness and response (*Moon et al., 2015*). In 2014 a zoonotic niche map for Ebola virus disease (EVD) was produced (*Pigott et al., 2014*) in part to assess the expected geographical extent of spillover risk. This research was then expanded to explore how changes in demography and international connectivity may have facilitated the establishment and rapid subsequent spread of the epidemic (*Bogoch et al., 2015*). The West African outbreak of EVD has again highlighted key information gaps that exist with respect to the broader epidemiology of Ebola virus, particularly concerning viral persistence in reservoirs (*Funk and Piot, 2014*; *Mari Saez et al., 2015*; *Leendertz, 2016*), and prompted a variety of questions concerning the role bats play in transmission (*Leendertz et al., 2016*). Identifying reservoirs of zoonotic disease is a complex process (*Viana et al., 2014*; *Haydon et al., 2002*) and whilst considerable sampling effort has been undertaken over the years (*Kuhn, 2008*; *Leirs et al., 1999*), isolation of Ebolavirus from living animals has been rare (*Leroy et al., 2005*). The original eLife study only incorporated the three bat species found to be RNA-positive (*Leroy et al., 2005*). Whilst this remains currently the best evidence for an animal reservoir species, it is important to consider that other sampling efforts may by chance represent false negatives, particularly if infection is rare.

Consequently, to contribute to these broader discussions, the original paper (*Pigott et al., 2014*) was updated with new occurrence data and expanded to consider a wider range of potential bat reservoir species. Bats remained the priority mammalian order given the previous viral isolation and the repeated anecdotal implications in previous outbreaks (*Leroy et al., 2009*; *Mari Saez et al., 2015*). Since there are a large number of bat species found in Africa, we defined three groupings, based upon the strength of evidence supporting their potential Ebola reservoir status. As a result, not only were the original three RNA-positive bats included (*Leroy et al., 2005*), but also those species with serological evidence of EVD infection (*Olival and Hayman, 2014*) and those identified through trait-based machine learning approaches as being similar to species already reporting filoviral infection (*Han et al., 2016*).

## Results

Six additional records of EVD were incorporated into the disease occurrence database: one human outbreak in the Democratic Republic of Congo (*Maganga et al., 2014*); two reports of infections in animals in Zambia (*Ogawa et al., 2015*); and three animal infections in Central African Republic (*Morvan et al., 1999*) (*Figure 1*). Of these new occurrences, two in southern Central African Republic are found in areas predicted to be at-risk by the previous model (*Pigott et al., 2014*), with the index case from the Democratic Republic of Congo located in close proximity (<10 km) to at-risk areas. The occurrences in Zambia and northern Central African Republic lie, respectively, to the south and north of previously predicted at-risk regions.

*Figure 2* depicts the three new consolidated bat distributions. The revised distribution of the Group 1 bats (*i.e.* those found to have been Ebolavirus RNA positive) is broadly consistent with that published in the original paper except that the peripheries of Central Africa are now predicted to be environmentally suitable for these bats, as well as some parts of East Africa, particularly Tanzania, Mozambique and Madagascar. The Group 2 and Group 3 bat species are predicted to be distributed across much of Africa stretching from West to East Sub-Saharan Africa, as well as much of the coastline of the continent.

The revised niche map, incorporating the updated bat covariates and disease occurrence database, is presented in *Figure 3*. The map shows the predicted areas of environmental suitability for zoonotic Ebola virus transmission to be consistent with previous attempts, but the relative environmental suitability within this distribution differs from the previous estimates. *Figure 3—figure supplement 1* demonstrates that Cameroon, Gabon, Republic of Congo and mainland Equatorial Guinea are now predicted to be more environmentally suitable than in the previous analysis. The regions of Central Africa (particularly Gabon and the Republic of Congo) identified as being most environmentally suitable for zoonotic EVD transmission in the previous analysis remain so in this analysis. The revised number of predicted at-risk countries, determined by thresholding the map by a probability that captures 95% of the occurrence dataset, is 23 (*Table 1*).

The similar AUC values ($0.85 \pm 0.04$ compared to $0.8236 \pm 0.080$) between the previous and current iterations suggest that the updated model fits the new occurrence dataset as well as the previous model fitted the older dataset. Mean enhanced vegetation index (EVI) remains the highest relative predictor covariate for zoonotic EVD transmission while the relative importance of Group 1 bat distributions moved from being the fifth most important to the second. Mean night-time land surface temperature (LST), elevation and mean daytime LST complete the top five predictors (*Table 2*).

When separate bat layers were used in the model, as opposed to the consolidated covariates, the predictions were geographically similar (*Figure 3—figure supplement 2*) however, four bat species were identified as explaining more of the variation than the rest; *Hypsignathus monstrosus*, *Epomops franqueti* (from both of which Ebolavirus RNA has been isolated), *Otomops martiensseni* and *Epomophorus labiatus* (both from Group 3). The explanatory power of this model (evaluated using AUC) was comparable to the model results described above (AUC = $0.819 \pm 0.080$).

## Discussion

This research advance integrates new data as well as a more thorough consideration of the bat species that act as a reservoir for the virus in order to update our modelled estimate of the zoonotic

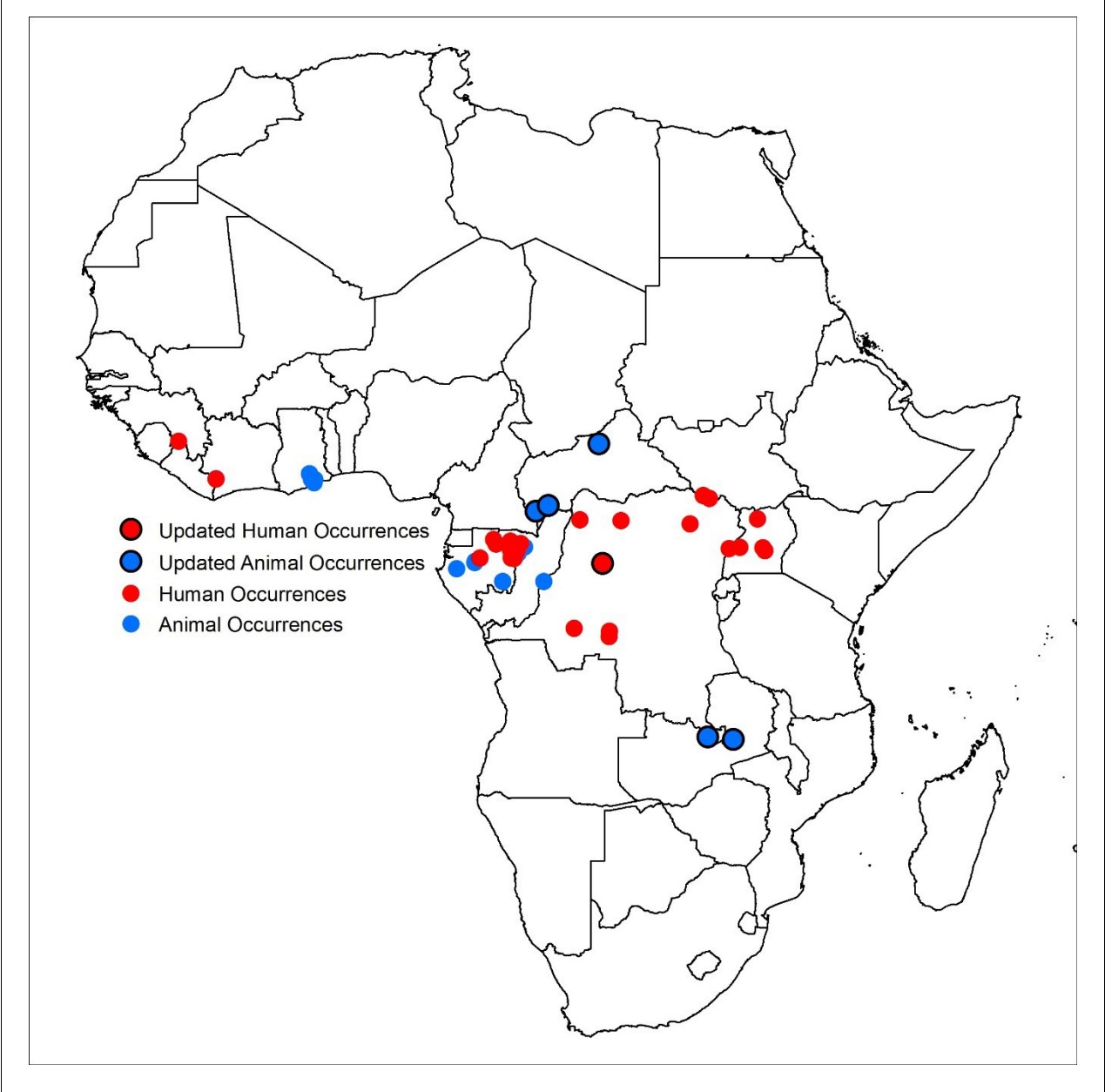

**Figure 1.** Updated Ebola virus disease occurrence database. Human index cases are represented by red circles, animal occurrences in blue. New occurrence information is indicated by the black circle. The coordinates of polygon centroids are displayed for occurrences defined by an area greater than 5 km x 5 km.

niche of EVD. The area estimated to be at potential risk of zoonotic EVD transmission has now expanded to include Kenya and the influence of additional bat species demonstrates that continued focus should be placed on rigorously identifying reservoir species and the role they play in sustaining viral transmission (*Leendertz, 2016*). The fact that *O. martiensseni* and *E. labiatus* contribute explanatory power to the model, in comparison with their distributions on the eastern and southern periphery of reported cases of EVD (*Figure 2—figure supplement 3*) suggests that different regions of the continent may support transmission cycles with differing reservoir species. This, coupled with the potential for each of the pathogenic species of *Ebolavirus* potentially having differing distributions (*Peterson et al., 2004*), cannot currently be explored more rigorously due to insufficient data.

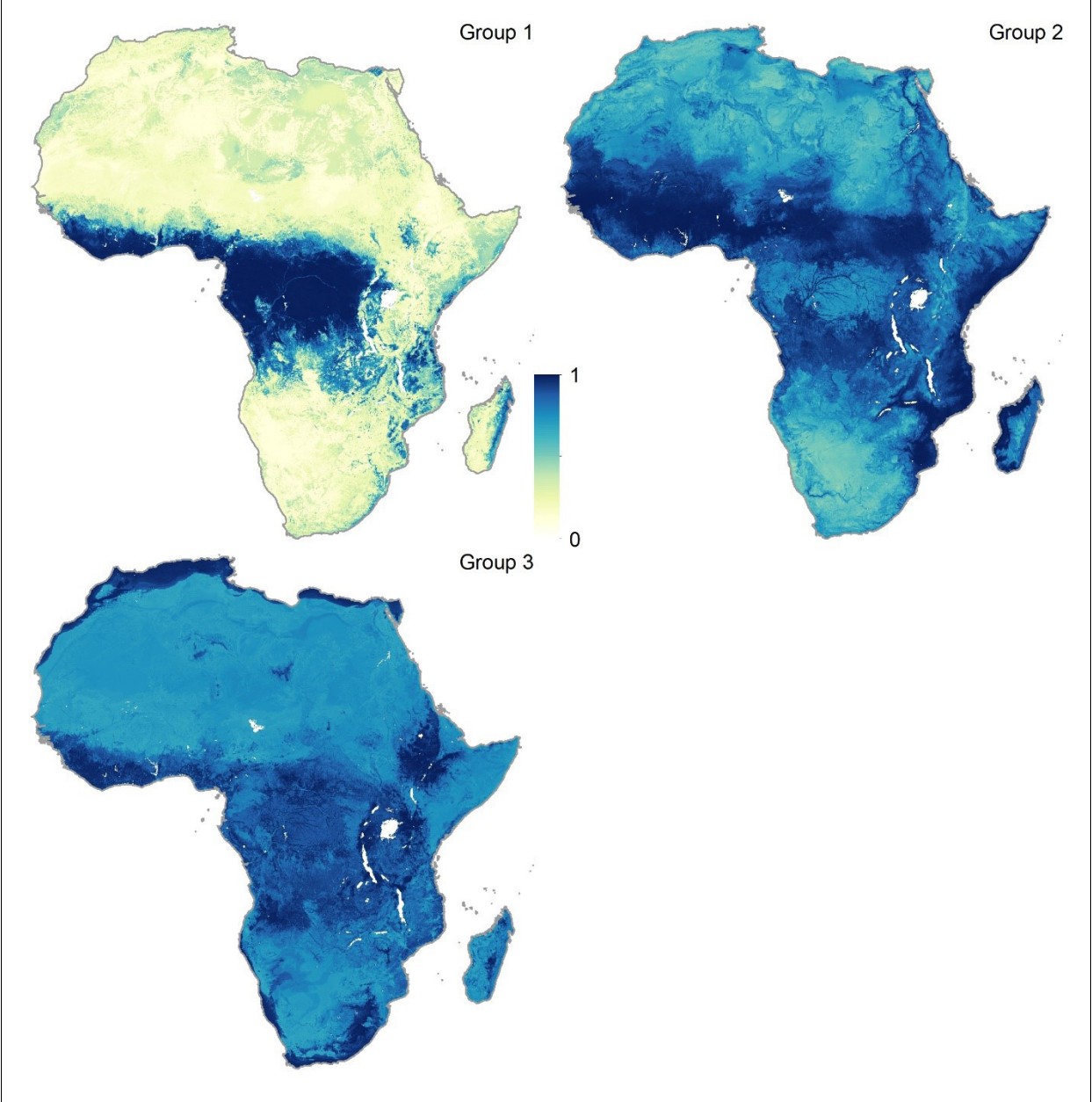

**Figure 2.** Combined suitability surfaces for each of the potential reservoir bat groupings. For each layer the species specific suitability maps were combined to produce a surface approximating the probability that any bat species in that group may be present. Regions in blue (1) are most environmentally similar to locations reporting bat records. Areas in yellow (0) are the least environmentally similar. The top left panel depicts Group 1, top right Group 2 and bottom left Group 3 bats.

The following figure supplements are available for figure 2:

**Figure supplement 1.** Group 1 bat distributions.

**Figure supplement 2.** Group 2 bat distributions.

**Figure supplement 3.** Group 3 bat distributions.

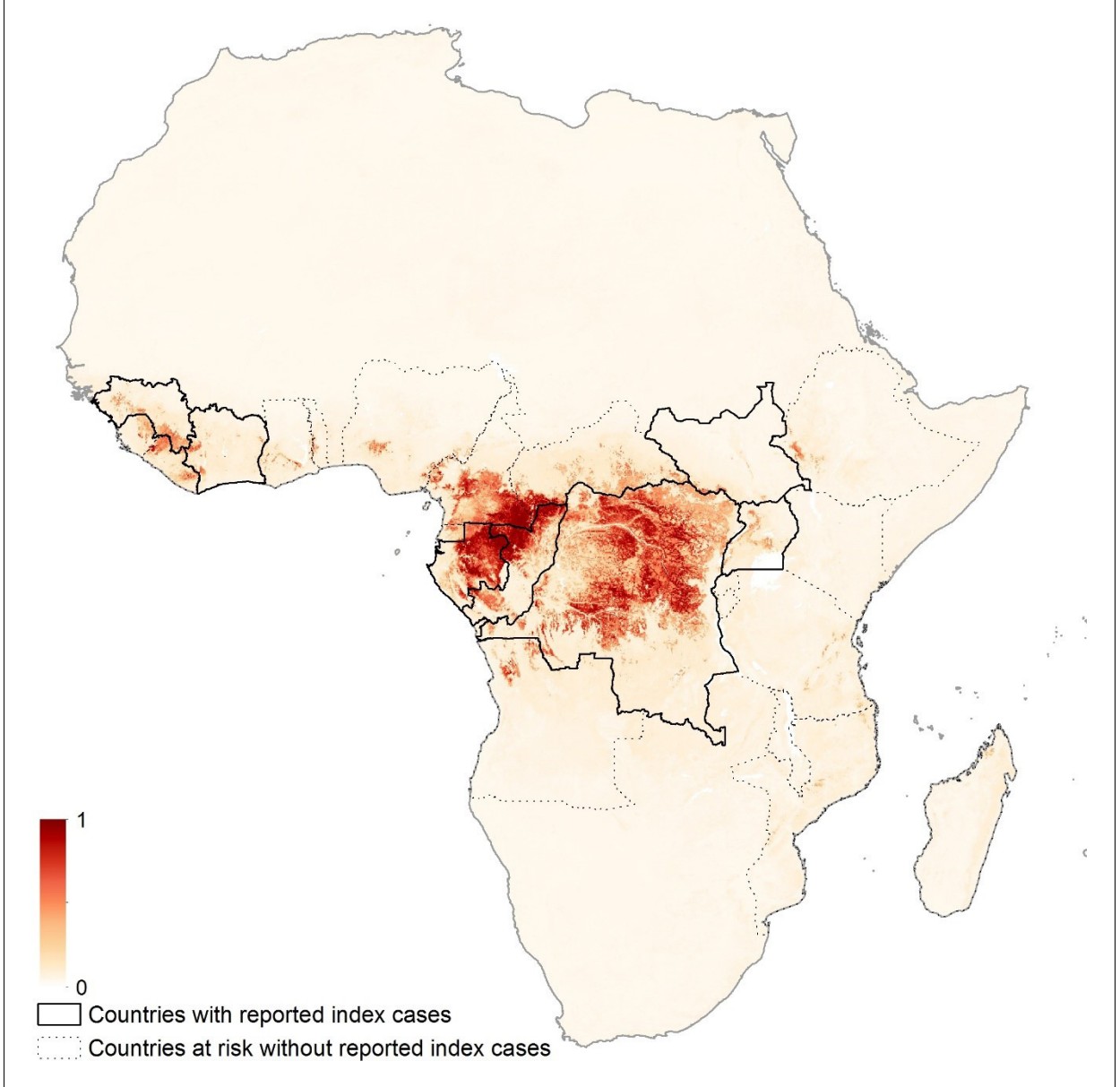

**Figure 3.** Updated map showing areas most environmentally suitable for the zoonotic transmission of Ebola virus. Areas closer to dark red (1) are most environmentally similar to locations reporting Ebola virus occurrences; areas in light yellow (0) are least similar. Countries with borders outlined are those which are predicted to contain at-risk areas for zoonotic transmission based on a thresholding approach. Output displayed generated from model using the three consolidated bat covariates.

The following figure supplements are available for figure 3:

**Figure supplement 1.** Absolute differences between previous and revised maps.

**Figure supplement 2.** Zoonotic niche map based upon inclusion of individual bat covariate layers.

As with the original publication, it must be reiterated that environmental suitability does not inevitably lead to spillover events. Currently absolute human population residing in at-risk pixels is used as a proxy for spillover likelihood, however, a variety of factors will influence the outbreak risk within each location (*Plowright et al., 2015*) and only by including covariates that consider human behaviour (*Woldehanna and Zimicki, 2015*), patterns of susceptibility in other animals (*Walsh et al.,*

**Table 1.** National populations at risk.

| | Country | Population-at-risk (in 100,000s) |
|---|---|---|
| Countries previously reporting index cases | Democratic Republic of the Congo | 170.18 |
| | Uganda | 21.58 |
| | Guinea | 17.61 |
| | Côte d'Ivoire | 4.08 |
| | Gabon | 3.65 |
| | South Sudan | 1.80 |
| | Republic of Congo | 1.07 |
| Countries with no reported index cases | Nigeria | 29.13 |
| | Cameroon | 22.90 |
| | Central African Republic | 7.62 |
| | Liberia | 5.88 |
| | Ghana | 4.04 |
| | Sierra Leone | 3.94 |
| | Angola | 3.25 |
| | Togo | 1.78 |
| | Ethiopia | 1.75 |
| | Equatorial Guinea | 1.22 |
| | Tanzania | 1.18 |
| | Burundi | 1.07 |
| | Mozambique | 0.55 |
| | Madagascar | <0.1 |
| | Kenya | <0.1 |
| | Malawi | <0.1 |

*2007*; *2009*), impacts of land use change (*Rosales-Chilama et al., 2015*) and within-host viral dynamics (*Amman et al., 2012*; *Hayman, 2015*) can an approximation of spillover risk be defined.

These updates demonstrate the ease and speed with which new data and covariate considerations can be incorporated within existing empirical models (*Kraemer et al., 2016*). As the wider discussion on EVD turns to focus on strategies to prevent or contain future spillover events as well as developing long-term in-country containment capacities (*Currie et al., 2016*), it is hoped that maps

**Table 2.** Comparison of previous and revised niche models.

| | Revised niche map (with summary bat layers) | Revised niche map (with individual bat layers) | Previous eLife niche map (*Pigott et al., 2014*) |
|---|---|---|---|
| AUC | 0.8236 ± 0.08 | 0.8195 ± 0.08 | 0.85 ± 0.04 |
| Occurrences | n = 57 (animals), n = 31 (humans) | n = 57 (animals), n = 31 (humans) | n = 51 (animals), n = 30 (humans) |
| Ranked relative contributions | EVI mean (0.55) | EVI mean (0.46) | EVI mean (0.65) |
| | Group 1 bat distribution (0.18) | *Hypsignathus monstrosus* (0.15) | Elevation (0.12) |
| | LST mean (night) (0.08) | *Epomops franqueti* (0.08) | LST mean (night) (0.08) |
| | Elevation (0.06) | *Otomops martiensseni* (0.06) | PET mean (0.06) |
| | LST mean (day) (0.04) | *Epomophorus labiatus* (0.04) | Bat distribution (0.04) |

such as these convey the heterogeneities in spillover risk that exist within Africa. To better enable researchers and policymakers to consider EVD preparedness and necessary contingencies, a new online tool has been developed which allows users to interrogate the revised maps in more detail, in areas of specific interest (http://vizhub.healthdata.org/ebola). As part of this tool, the zoonotic niche output and Group 1 bat layers are now available, along with filters for identifying at-risk countries and locations of previous index cases from outbreaks.

Geographic datasets such as these provide context to broader discussions as our aspirations transition from controlling outbreaks to mitigating the risk of future spillover events prioritised by their potential for more widespread epidemics. Such data are particularly important for determining where best to investigate the frequency of potentially transmissible contacts between reservoir and susceptible species and humans. Previous niche maps served as an important impetus in the search for potential reservoirs (*Peterson et al., 2004*) and these iterations can continue to inform such work. As researchers and policy makers seek to resolve outstanding questions about EVD epidemiology, it is hoped that the continued updating and dissemination of this information can contribute to this discussion.

## Materials and methods

### Updating the occurrence database

Since the previous publication, an outbreak of EVD occurred in humans in the Democratic Republic of the Congo (*Maganga et al., 2014*). The outbreak is thought to have originated in Inkanamongo, a village near Boende, Équateur province and resulted in 66 probable and confirmed cases and 49 probable and confirmed deaths (*Rosello et al., 2015*). A polygon of radius 10 km centered on the town of Boende was included to capture the location of the index case, increasing the database of assumed independent animal-to-human spillover events to 31 as part of 24 distinct reported outbreaks (*Mylne et al., 2014*).

In addition, a re-analysis of the literature available on infections in animal species was completed on 7th October 2015. Due to the poor differential capacity of immunological tests to discriminate Ebola virus from other viruses we retained the following inclusion criteria for the database; for susceptible species mortality events linked to Ebolavirus by any diagnostic methodology *or* PCR-positive diagnosis of Ebolavirus were included. Inclusion criteria for potential bat reservoir species were either PCR-positivity or serological evidence suggesting Ebolavirus infection. Serological studies were included without fatal outcomes (unlike with susceptible species) due to the hypothesised asymptomatic nature of infection in the reservoir hosts. As a result of these inclusion criteria, studies with serological detection of Ebolavirus in healthy non-Chiropteran species were excluded, such as surveys in dogs (*Allela et al., 2005*).

In total six new records of EVD occurrence in animals were identified and included within the database to increase the total to 57. These records were obtained from two research articles. The first of these assessed Ebolavirus load in a variety of mammal species and identified PCR-positivity in a number of small mammals across three sites in Central African Republic (*Morvan et al., 1999*). In total, four separate occurrences, consisting of three different species, were identified as being PCR positive: a member of the *Praomys* complex, Peter's mouse (*Mus setulosus*) and the greater forest shrew (*Sylvisorex ollula*). The second study investigated serological responses in straw-coloured fruit bats (*Eidolon helvum*) caught in two districts in Zambia (*Ogawa et al., 2015*). Specific latitudes and longitudes of the study sites were supplied for the Central African Republic study and were used to generate point occurrences. For the Zambian study it was necessary to use administrative data representing the two districts where the bats were caught (Serenje and Ndola districts).

### Expanding potential bat reservoir species

Potential bat reservoir species were stratified into three groupings based upon the strength of evidence suggesting their reservoir status (*Table 3*). Group 1 contained the three species of bat from which Ebolavirus RNA has been detected and therefore have the strongest evidence to support potential reservoir status (*Leroy et al., 2005*). Group 2 species are those that, using a variety of serological tests, have been reported to be Ebolavirus seropositive, suggesting potential reservoir status. A previous review (*Olival and Hayman, 2014*), identified nine species as seropositive for

**Table 3.** Final bats included in analysis classified by evidence grouping.

| Grouping | Bat | Occurrences |
| --- | --- | --- |
| Group 1 | Franquet's epauletted fruit bat (*Epomops franqueti*) | 442 |
| | Hammerheaded fruit bat (*Hypsignathus monstrosus*) | 254 |
| | Little collared fruit bat (*Myonycteris torquata*) | 107 |
| Group 2 | Angolan free-tailed bat (*Tadarida condylura*, formerly *Mops condylurus*) | 179 |
| | Egyptian fruit bat (*Rousettus aegyptiacus*) | 177 |
| | Gambian epauletted fruit bat (*Epomophorus gambianus*) | 166 |
| | Peter's dwarf epauletted fruit bat (*Micropteropus pusillus*) | 208 |
| | Straw-coloured fruit bat (*Eidolon helvum*) | 282 |
| Group 3 | Buettikofer's epauletted fruit bat (*Epomops buettikoferi*) | 50 |
| | Common bent-wing bat (*Miniopterus schreibersii*) | 31 |
| | Eloquent horseshoe bat (*Rhinolophus eloquens*) | 61 |
| | Ethiopian epauletted fruit bat (*Epomophorus labiatus*) | 187 |
| | Giant leaf-nosed bat (*Hipposideros gigas*) | 21 |
| | Greater long-fingered bat (*Miniopterus inflatus*) | 56 |
| | Large-eared free-tailed bat (*Otomops martiensseni*) | 33 |

Ebolavirus. This was reduced to five species after the removal of the three species already categorised in Group 1 and Leschenault's Rousette, *Rousettus leschenaultii*, which is not found in Africa.

Finally, Group 3 species were identified *via* generalized boosted regression analysis, which discriminates the bats reported to be filovirus-positive by learning trait patterns that distinguish them from all other bat species (*Han et al., 2016*). Generalized boosted regression (*Elith et al., 2008*) was applied to traits describing all bat species, including life history, physiological, ecological, morphological and demographic variables collected from numerous published sources. In addition to traits, the filovirus status of each bat species was assigned as a binary score (0 – not currently known to be positive for any filoviruses; 1 – published evidence). This analysis produces a rank list of all bat species according to their probability of being a filovirus carrier on the basis of their trait similarities with known filovirus-positive bat species. Bats found in the 90[th] percentile of likely filovirus carriers were initially considered, and then filtered to include only those which have home ranges in Africa (*Schipper et al., 2008*). As per the original publication, occurrence records were extracted from the Global Biodiversity Information Facility (GBIF). Species for which there were fewer than 20 unique GBIF records in Africa were dropped from the analysis due to data paucity. *Table 3* reports the bat species and corresponding numbers of occurrences included in the analysis.

For Group 1 species, occurrence records were supplemented by searching PubMed and Web of Knowledge for additional reports. A literature review was completed on the 8[th] September 2015 using the following sets of keywords:

- '*Hypsignathus monstrosus*' or 'hammer-headed bat' or 'hammer headed bat' or 'hammer-headed bat' or 'big-lipped bat' or 'big lipped bat' or '*Hypsignathus labrosus*' or '*Hypsignathus macrocephalus*'
- '*Myonycteris torquata*' or 'little collared fruit bat' or '*Myonycteris collaris*' or '*Myonycteris leptodon*' or '*Myonycteris wroughtoni*'
- '*Epomops franqueti*' or "Franquet's epauletted fruit bat" or '*Epomops comptus*' or '*Epomops strepitans*'

A total of 34 articles were identified for inclusion, from which 564 additional occurrences were sourced.

All bat species were modelled separately using boosted regression trees (*Elith et al., 2008*) utilising the same modelling procedure as outlined in the original article except that 100, rather than 50, bootstrap models were fitted. This resulted in 15 individual environmental suitability maps for bat species (see *Figure 2—figure supplements 1*, *2* and *3*), as well as three consolidated bat layers combining the environmental suitability maps for the bats within each of the three groupings (*Figure 2*).

## Revising the predicted zoonotic niche map

A species distribution model, specifically a boosted regression trees approach (*Elith et al., 2008*), was implemented. The model generates regression trees based upon binary splits of linked covariates, which are iteratively improved upon by boosting. The regression trees are capable of characterising complex environmental interactions and correlations since each tree is built from a hierarchy of multiple nodes, each based upon different successive binary splits of the covariates. The model extracts environmental information for each reported occurrence of Ebolavirus to define an optimal relationship between presence of the disease and environmental factors. Predictive performance is improved by including a comparison background dataset that acts as a hypothesised environmental negative control (*Phillips et al., 2009*). As per the previous analysis, this dataset was generated by randomly sampling across Africa biased towards areas of high population density. By including human population density in this way, some potential sampling biases present in human index case reporting can be mitigated as cases are more likely to be reported in more populous areas. The boosted regression trees were re-run using the same parameters and covariates (elevation, mean evapotranspiration rate, and mean and range measures of enhanced vegetation index, daytime land surface temperature (LST), and night-time LST) as the previous publication except for the inclusion of the new occurrence data outlined above and the new bat layers. Two model iterations were run: one with the three consolidated bat layers (i.e. Groups 1, 2 and 3) and the other with all the bat species layers considered separately.

## Estimating populations at risk

The continuous suitability surface was converted into a binary at-risk versus not-at-risk surface by determining a threshold value that included 95% of the estimated suitability values of pixels with reported human index cases (*Pigott et al., 2015*). For sites represented by a specific latitude and longitude the suitability score was taken from the corresponding pixel; for polygon estimates covering a number of cells, the mean suitability was taken across all pixels covered by the polygon.

Whilst not included directly as a covariate in the modelling process, human population layers were assessed in at-risk locations as a potential proxy for spillover frequency. The populations living within the gridded cells thought to be at-risk of potential Ebolavirus transmission from zoonotic sources were calculated using an updated contemporary gridded estimate of population (*WorldPop Project, 2015*).

## Acknowledgements

This work was funded by a grant from The Paul G. Allen Family Foundation (#11878) to SIH which also provided funding for DMP, AIM, LE and CM. BAH acknowledges support from the National Institute of General Medical Sciences of the National Institutes of Health (#U01GM110744). FMS is funded by the Rhodes Trust. OJB is funded by the Bill & Melinda Gates Foundation (#OPP1119467). MUGK is funded by the German Academic Exchange Service (DAAD) through a graduate scholarship. CLM and NG acknowledge support from the Bill & Melinda Gates Foundation (#OPP1053338).

PWG is a Career Development Fellow (#K00669X) jointly funded by the UK Medical Research Council (MRC) and the UK Department for International Development (DFID) under the MRC/DFID Concordat agreement and receives support from the Bill & Melinda Gates Foundation (#OPP1068048, #OPP1106023) which also supports DJW and SB. SIH is also funded by a Senior Research Fellowship from the Wellcome Trust (#095066) and grants from the Bill & Melinda Gates Foundation (#OPP1093011, #OPP1132415). DMP and BAH also acknowledge the support for a collaborative meeting from the RAPIDD program of the Science & Technology Directorate. SIH also acknowledges a grant from the Research for Health in Humanitarian Crises (R2HC) Programme, managed by ELHRA (#13468) and funded equally by the Wellcome Trust and DFID, which also supported NG and MUGK.

## Additional information

### Competing interests

SIH: Reviewing editor, *eLife*. The other authors declare that no competing interests exist.

### Funding

| Funder | Grant reference number | Author |
|---|---|---|
| Paul G. Allen Family Foundation | 11878 | David M Pigott<br>Anoushka I Millear<br>Lucas Earl<br>Chloe Morozoff<br>Simon I Hay |
| Science and Technology Directorate | RAPIDD program | David M Pigott<br>Barbara A Han |
| National Institute of General Medical Sciences | U01GM110744 | Barbara A Han |
| Rhodes Scholarships | | Freya M Shearer |
| Bill and Melinda Gates Foundation | OPP1068048 | Daniel J Weiss<br>Samir Bhatt<br>Peter W Gething |
| Bill and Melinda Gates Foundation | OPP1106023 | Daniel J Weiss<br>Samir Bhatt<br>Peter W Gething |
| Bill and Melinda Gates Foundation | OPP1119467 | Oliver J Brady |
| German Academic Exchange Service London | Graduate Scholarship | Moritz UG Kraemer |
| Research for Health in Humanitarian Crises (R2HC) Programme | ELHRA (#13468) | Moritz UG Kraemer<br>Nick Golding<br>Simon I Hay |
| Wellcome Trust | | Moritz UG Kraemer<br>Nick Golding |
| Department for International Development | | Moritz UG Kraemer<br>Nick Golding |
| Bill and Melinda Gates Foundation | OPP1053338 | Catherine L Moyes<br>Nick Golding |
| Medical Research Council | MRC/DFID Concordat agreement | Peter W Gething |
| Department for International Development | MRC/DFID Concordat agreement | Peter W Gething |
| Wellcome Trust | Senior Research Fellowship #095066 | Simon I Hay |
| Bill and Melinda Gates Foundation | OPP1093011 | Simon I Hay |

| | | |
|---|---|---|
| Bill and Melinda Gates Foundation | OPP1132415 | Simon I Hay |

The funders had no role in study design, data collection and interpretation, or the decision to submit the work for publication.

## Author contributions

DMP, SIH, Conception and design, Acquisition of data, Analysis and interpretation of data, Drafting or revising the article; AIM, BAH, Acquisition of data, Analysis and interpretation of data, Drafting or revising the article; LE, CM, FMS, OJB, MUGK, Analysis and interpretation of data, Drafting or revising the article; DJW, Acquisition of data, Drafting or revising the article; CLM, NG, Conception and design, Analysis and interpretation of data, Drafting or revising the article; SB, PWG, Conception and design, Drafting or revising the article

## Author ORCIDs

Chloe Morozoff, http://orcid.org/0000-0002-3254-5553
Moritz UG Kraemer, http://orcid.org/0000-0001-8838-7147
Catherine L Moyes, http://orcid.org/0000-0002-8028-4079
Simon I Hay, http://orcid.org/0000-0002-0611-7272

# Additional files

## Major datasets

The following dataset was generated:

| Author(s) | Year | Dataset title | Dataset URL | Database, license, and accessibility information |
|---|---|---|---|---|
| David M Pigott, Anoushka I Millear, Lucas Earl, Chloe Morozoff, Barbara A Han, Freya M Shearer, Daniel J Weiss, Oliver J Brady, Moritz UG Kraemer, Catherine L Moyes, Samir Bhatt, Peter W Gething, Nick Golding, Simon I Hay | 2016 | eLife Advances - Updates to the zoonotic niche map of Ebola virus disease in Africa | http://dx.doi.org/10.6084/m9.figshare.c.3267965.v1 | Publicly available at figshare (https://figshare.com/) |

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
