## [Decision Letter]

Thank you for submitting your article "Updates to the zoonotic niche map of Ebola virus disease in Africa" for consideration by *eLife*. Your article has been reviewed by Pasi Penttinen and Armand Sprecher, and the evaluation has been overseen by a Reviewing Editor and Prabhat Jha as the Senior Editor.

The reviewers have discussed the reviews with one another and the Reviewing Editor has drafted this decision to help you prepare a revised submission.

Summary:

The reviewers and Reviewing Editor felt that this update to your previous model on the zoonotic niche for Ebola virus disease in Africa (Pigott et al. 2014) addressed an important, original and timely research question. The study was well designed, analysed and presented. The manuscript is clearly written in good English, and was an interesting read.

While the reviewers supported the general idea of updating previous maps to incorporate new data, there were reservations in particular about the inclusion of bat groups 2 and 3 and the clinical inclusion of animal cases. We would like to see a revised manuscript with much more explanation and support for these inclusions (as detailed in major comment 1 below).

Essential revisions:

1) The explanatory power from the two bats species in group 3 was interesting. However, it felt like a step had been omitted. The "more thorough consideration of the bat species that act as a reservoir for the virus" is a good idea, but it would seem one that requires being established first prior to incorporation into the map. The addition of groups 2 and 3 seems too casual here. For example, the paper admits in the Materials and Materials and methods section "the poor differential capacity of immunological tests to discriminate Ebola virus from other viruses" is such that weak clinical criteria need be used to identify infections in animal species, and then notes that the inclusion criterion for bat reservoir species were "serological evidence suggesting Ebolavirus infection". Furthermore, it is not clear how the nine bat species listed in Table 1 of the Olival and Hayman article cited became the five species in Group 2. It is hard then for the reader to get very excited about the bats in Group 2 without more support.

For Group 3, the reader is even more at sea, as the referenced article is "in review", and all we know is that some machine learning routine sorted out some bat species according to likelihood of filovirus carriage somehow. That two species produced interesting explanatory power is interesting, but they have been drawn from a rather black box.

Hence we feel that there needs to be a much more thorough consideration of the bat species for the incorporation of these additions to be justified.

2) Coming back to the criteria for inclusion of animal cases, the previous niche map seems to have used only those cases identified by serology and PCR. The current manuscript adds mortality and other clinical symptoms (subsection “Updating the occurrence database”, second paragraph), noting this is consistent with inclusion criteria for humans. Importantly, the clinical inclusion criteria for humans are only used in the epidemic zone of a laboratory confirmed outbreak, and this is only for identification of suspect cases who will undergo subsequent laboratory testing (or if this is impossible, epidemiologic criteria may be used to identify probable cases). Ebola is a vague disease in humans, and may be nonspecific in animals. The clinical criteria the authors use are not given, and it is unclear how specific they might be. More information is needed on this to evaluate the suitability of inclusion on clinical grounds.

3) You do not present or discuss how well the newly identified EVD events fit into their previous model. Please consider adding a layer with the previously published model (Figure 5B? from the 2014 publication) on Figure 1 and/ or discuss how the newly identified events fit with the previous model.

4) Spatial distribution of the human population is not incorporated in the model. It would be useful for more generalist readers to describe briefly how the boosted regression tree modelling takes (or not) into account the fact that it is quite possible that the human population density and environmental variables (including bat population density) are correlated. Hence, a fit with occurrences of zoonotic events might be partly a reflection of population density.

---

## [Author Response]

*Summary:*

The reviewers and Reviewing Editor felt that this update to your previous model onthe zoonotic niche for Ebola virus disease in Africa (Pigott et al. 2014)addressed an important, original and timely research question. The study was welldesigned, analysed and presented. The manuscript is clearly written in goodEnglish, and was an interesting read.

While the reviewers supported the general idea of updating previous maps toincorporate new data, there were reservations in particular about the inclusion ofbat groups 2 and 3 and the clinical inclusion of animal cases. We would like to see a revised manuscript with much more explanation and support for these inclusions (as detailed in major comment 1 below).

We thank the editors and reviewers for the overall positive comments. We havehopefully added sufficient details (outlined below) to respond to the concernsabout justifying additional reservoir species.

*Essential revisions:*

*1) The explanatory power from the two bats species in group 3 was interesting. However, it felt like a step had been omitted. The "more thorough consideration of the bat species that act as a reservoir for the virus" is a good idea, but it would seem one that requires being established first prior to incorporation into the map. The addition of groups 2 and 3 seems too casual here. For example, the paper admits in the Materials and methods section "the poor differential capacity of immunological tests to discriminate Ebola virus from other viruses" is such that weak clinical criteria need be used to identify infections in animal species, and then notes that the inclusion criterion for bat reservoir species were "serological evidence suggesting Ebolavirus infection".*

We have added additional text providing background information on potential reservoir host selection processes and why it is important, as well as discussing this within the context of Ebola virus disease (EVD). We have added the following text to the Introduction:

“Identifying reservoirs of zoonotic disease is a complex process (Viana et al. 2014, Haydon et al. 2002) and whilst considerable sampling effort has been undertaken over the years (Kuhn 2008; Leirs et al. 1999), isolation of Ebolavirus from living animals has been rare (Leroy et al. 2005). […] Since there are a large number of bat species found in Africa, we defined three groupings, based upon the strength of evidence supporting their potential Ebola reservoir status. As a result, […]”

Furthermore, it is not clear how the nine bat species listed in Table 1 of the Olival and Hayman article cited became the five species in Group 2. It is hard then for the reader to get very excited about the bats in Group 2 without more support.

Olival and Hayman’s review identifies nine species of bat as being positive for Zaire Ebolavirus by either serology or PCR. Of these nine, three (*Epomops franqueti*, *Hypsignathus monstrosus* and *Myonycteris torquata*) are already considered within Group 1 due to the PCR identification. Of the remaining six, one (*Rousettus leschenaultii*) is only found in Asia and therefore was excluded due to being out of the study geography. While these species may not be classified via gold-standard methods such as PCR, they represent a distinct grouping of potential reservoir species in comparison to other bat species (where no serological evidence has been found). We have added the following to the text to the Materials and methods section:

“Group 2 series are those that, using a variety of serological tests, have been reported to be Ebolavirus seropositive, suggesting potential reservoir status. A previous review (Olival & Hayman 2014), identified nine species as seropositive for Ebolavirus. This was reduced to five species after the removal of the three species already categorized in Group 1 and Leschenault’s Rousette, *Rousettus leschenaultii*, which is not found in Africa.”

*For Group 3, the reader is even more at sea, as the referenced article is "in review", and all we know is that some machine learning routine sorted out some bat species according to likelihood of filovirus carriage somehow. That two species produced interesting explanatory power is interesting, but they have been drawn from a rather black box.*

Hence we feel that there needs to be a much more thorough consideration of the bat species for the incorporation of these additions to be justified.

We have considerably bolstered this section and added the following text:

“Finally, Group 3 species were identified via generalized boosted regression analysis, which discriminates the bats reported to be filovirus-positive by learning trait patterns that distinguish them from all other bat species (Han et al. 2016). […] This analysis produces a rank list of all bat species according to their probability of being a filovirus carrier on the basis of their trait similarities with known filovirus-positive bat species. Bats found in the 90^th^ percentile of likely filovirus […]”

2) Coming back to the criteria for inclusion of animal cases, the previous niche map seems to have used only those cases identified by serology and PCR. The current manuscript adds mortality and other clinical symptoms (subsection “Updating the occurrence database”, second paragraph), noting this is consistent with inclusion criteria for humans. Importantly, the clinical inclusion criteria for humans are only used in the epidemic zone of a laboratory confirmed outbreak, and this is only for identification of suspect cases who will undergo subsequent laboratory testing (or if this is impossible, epidemiologic criteria may be used to identify probable cases). Ebola is a vague disease in humans, and may be nonspecific in animals. The clinical criteria the authors use are not given, and it is unclear how specific they might be. More information is needed on this to evaluate the suitability of inclusion on clinical grounds.

We have clarified this section. No clinical symptoms were reported in animals beyond fatal outcomes. To avoid confusion, comparisons with the human inclusion criteria and references to clinical symptoms were dropped. We have also added an example of a seropositive animal survey that was excluded due to these criteria. The following sentences are now in the methodology:

“for susceptible species mortality events linked to Ebolavirus by any diagnostic methodology or […]”

“Serological studies were included without fatal outcomes (unlike with susceptible species) due to the hypothesized asymptomatic nature of infection in the reservoir hosts. As a result of these inclusion criteria, studies with serological detection of Ebolavirus in healthy non-Chiropteran species were excluded, such as surveys in dogs (Allela et al. 2005).”

3) You do not present or discuss how well the newly identified EVD events fit into their previous model. Please consider adding a layer with the previously published model (Figure 5B? from the 2014 publication) on Figure 1 and/ or discuss how the newly identified events fit with the previous model.

We thank the reviewers for suggesting this additional point for discussion. We have added the following within the Results section discussing how these points relate to the previous model:

“Of these new occurrences, two in southern Central African Republic are found in areas predicted to be at-risk by the previous model (Pigott et al. 2014), with the index case from the Democratic Republic of Congo located in close proximity (<10km) to at-risk areas. The occurrences in Zambia and northern Central African Republic lie, respectively, to the south and north of previously predicted at-risk regions.”

*4) Spatial distribution of the human population is not incorporated in the model. It would be useful for more generalist readers to describe briefly how the boosted regression tree modelling takes (or not) into account the fact that it is quite possible that the human population density and environmental variables (including bat population density) are correlated. Hence, a fit with occurrences of zoonotic events might be partly a reflection of population density.*

The boosted regression tree (BRT) model does not incorporate human population density as a covariate. This was done since human population density is unlikely to directly influence the zoonotic transmission of the virus; human index cases likely represent the tip of the iceberg of more widespread viral transmission. That said, human population density will likely influence any detection bias in human cases reported. This potential effect is reflected by using human population to bias the random selection of background or control data points required by the BRT modelling process. Human population density is then used to characterize the populations living in areas at risk, reflecting the fact that whilst some countries have relatively small geographic risk, they are relatively densely populated – this is highlighted by comparing the populations at risk between Gabon (large geographic area, lower total population) versusNigeria (smaller geographic area, higher total population). We have added two text sections, the first clarifying human population usage in the Materials and methods section and the second looking at the use of absolute population to define risk in the Discussion. We have also fleshed out the modelling section of the methods to clarify which covariates are included and how ecological interaction is dealt with in general:

(in Discussion): “Currently absolute human population residing in at-risk pixels is used as a proxy for spillover likelihood, however a variety of […]”

(in Materials and methods): “A species distribution model, specifically a boosted regression trees approach (Elith et al. 2008), was implemented. […] The boosted regression trees were re-run using the same parameters and covariates (elevation, mean evapotranspiration rate, and mean and range measures of enhanced vegetation index, daytime land surface temperature (LST), and night-time LST) as the previous […]”